# Similarity and Froude Number Similitude in Kinematic and Hydrodynamic Features of Solitary Waves over Horizontal Bed

**Chang Lin [1,\*], Ming-Jer Kao [1], James Yang [2,3], Rajkumar Venkatesh Raikar [4], Juan-Ming Yuan [5] and Shih-Chun Hsieh [6]**

[1] Department of Civil Engineering, National Chung Hsing University, Taichung 40227, Taiwan; mjkao1976@gmail.com
[2] Vattenfall AB, R&D Hydraulic Laboratory, 81470 Älvkarleby, Sweden; james.yang@vattenfall.com
[3] Department of Civil and Architectural Engineering, Royal Institute of Technology, 10044 Stockholm, Sweden
[4] Department of Civil Engineering, KLE Dr. M.S. Sheshgiri College of Engineering and Technology, Udyambag, Belagavi 590008, Karnataka, India; rvraikar@gmail.com
[5] Department of Data Science and Big Data Analytics, Providence University, Taichung 43301, Taiwan; jmyuan@pu.edu.tw
[6] Axesea Engineering Technology Co., Ltd., Taichung 407034, Taiwan; iraqhsc@gmail.com
\* Correspondence: chenglin@nchu.edu.tw

**Abstract:** This study presents, experimentally, similarity and Froude number similitude (FNS) in the dimensionless features of two solitary waves propagating over a horizontal bed, using two wave gauges and a high-speed particle image velocimetry (HSPIV). The two waves have distinct wave heights $H_0$ (2.9 and 5.8 cm) and still water depths $h_0$ (8.0 and 16.0 cm) but identical $H_0/h_0$ (0.363). Together with the geometric features of free surface elevation and wavelength, the kinematic characteristics of horizontal and vertical velocities, as well as wave celerity, are elucidated. Illustration of the hydrodynamic features of local and convective accelerations are also made in this study. Both similarity and FNS hold true for the dimensionless free surface elevation (FSE), wavelength and celerity, horizontal and vertical velocities, and local and convective accelerations in the horizontal and vertical directions. The similarities and FNSs indicate that gravity dominates and governs the wave kinematics and hydrodynamics.

**Keywords:** solitary wave; HSPIV; similarity; Froude number similitude; velocity; acceleration

## 1. Introduction

In the field, observations on the movement of solitary waves over a long distance were first reported by Russell [1], who also conducted laboratory experiments and derived an empirical formula for wave celerity. A solitary wave travels steadily in the wave-propagation direction and exhibits a stable state of motion. In other words, the wave maintains nearly constant wavelength with a slight decrease in wave height (Keulegan [2]), thus is regarded as one type of permanent wave. The solitary wave resembles a long wave because of its feature underlying the free surface of wave traveling over shallow water. Therefore, the study on solitary waves is of interest because of their simple and permanent wave form. Investigations on characteristics of flow velocity and acceleration contribute to the understanding of long wave kinematics and hydrodynamics (Liu et al. [3]; Hsiao et al. [4]; Higuera et al. [5]; Lin et al. [6–15]).

For solitary waves propagating over a *horizontal bed* at constant water depth, different methods were used for prediction of their wave profiles and associated fluid particle velocities. These include the analytical solution derived by Boussinesq [16]; the theoretical solution proposed by McCowan [17], Munk [18], Grimshaw [19], Fenton [20], Synolakis [21], Liu et al. [22], and Gavrilyuk et al. [23]; and the numerical results performed by Higuera et al. [5]. However, there are limited experimental studies that focus on the characteristics of the time series or profiles of horizontal and vertical velocities for solitary

wave traveling on a horizontal bed (e.g., Hsiao et al. [4], Lin et al. [8,15], and Lee et al. [24]). Further, Lin et al. [15] also elucidated temporal and spatial features of accelerations and pressure gradients in the horizontal and vertical directions.

Many experimental studies have been made on time series or profiles of free surface elevation (FSE), velocities, accelerations, or pressure gradients of solitary waves propagating over a *sloping bed* with run-up and run-down motions. Typical examples include Hsiao et al. [4], Lin et al. [6,7,9,10,14], Hall and Watts [25], Saeki et al. [26], Zelt [27], Briggs et al. [28], Jensen et al. [29], Fuchs and Hager [30], Pedersen et al. [31], Salevic et al. [32], and Smith et al. [33]. It is expected that the results of these physical hydraulic models should represent most parts of scenarios in a real-world prototype. However, differences between model and prototype parameters might exist due to a specified model used. Together with the scale and measurement tool adopted in the experiments, the distinctions are influenced by the model law used (e.g., Froude, Reynolds, or Weber number similitude). Among the previous studies mentioned above, Fuchs and Hager [30] was the first to elucidate the scale effect of solitary waves on the maximum run-up heights over a 1:5 sloping bed. Moreover, Lin et al. [14] investigated whether or not the model scale affects the flow features near the maximum run-up heights of solitary waves over a 1:3 sloping bed. Both studies indicated that, for experimental cases at $h_0$ no less than 8.0 cm, similarities exist not only in the dimensionless maximum run-up height but also in the dimensionless time (at which run-up motion of the wave tip ends). Lin et al. [14] also addressed similarity or non-similarity in the fine structure of swash tip and contact point subject to complex interplay among gravity, viscous friction, and surface tension.

To date, no related study has been carried out on the topic of similarity and/or FNS for solitary waves traveling on a *horizontal bed*. Due to lack of associated literature, the results from small-scale models, especially at $h_0 \leq 8.0$ cm, were frequently questioned regarding their small water depths (e.g., Higuera et al. [5], Lin et al. [6–10,15], and Watanabe and Horii [34]). It is with anticipation that, for small-scale experiments, the effects of viscous friction and/or surface tension may play certain roles in external and internal behaviors of solitary waves. This situation thus leads to the query about their influence and representativeness in data presentation. To shed light on the issue, this study addresses, in dimensionless form, similarity and Froude number similitude (FNS) in the FSEs, velocities, and accelerations for two solitary waves traveling over a horizontal bed at distinct length scales (i.e., $h_0 = 8.0$ and 16.0 cm).

The paper is outlined as follows. Section 2 illustrates the experimental setup and instrumentation, followed by Section 3 with preliminary tests. An introduction of FNS is briefed in Section 4. Detailed results and discussions with similarity and FNS are presented in Section 5. Finally, the findings are summarized with conclusions.

## 2. Experimental Setup

### 2.1. Wave Flume

Experiments were performed in a wave flume, 14.00 m long, 0.25 m wide, and 0.50 m high. Its bottom and two sidewalls were all made of glass plates. A precision servo-motor actuated a piston-type wave maker, allowing its movements to fully follow the waveplate trajectory developed by Goring [35]. A satisfactory solitary wave was thus generated in each run with high repeatability and nearly without the dispersive tail-wave, as reported by Lin et al. [6–10,14,15].

As seen in Figure 1, an $(x, y)$ coordinate system is defined with its origin $(0, 0)$ on the surface of a horizontal bed. The $x$ axis is oriented horizontally in the wave-propagation direction; and the $y$ axis is normal to the surface of the bed (positive upwards). The specified measurement section at $x = 0$ (denoted as SMS) is at an 800.0 cm distance from the wave maker at rest. Herein, a dimensionless time is defined as $T = t \times (g/h_0)^{1/2}$. Time $t = 0$ or $T = 0$ corresponds to the instant when the wave crest passes the SMS. The velocity components in the $(x, y)$ directions are denoted as horizontal and vertical velocities, $(u, v) = (u[x, y, t], v[x, y, t])$.

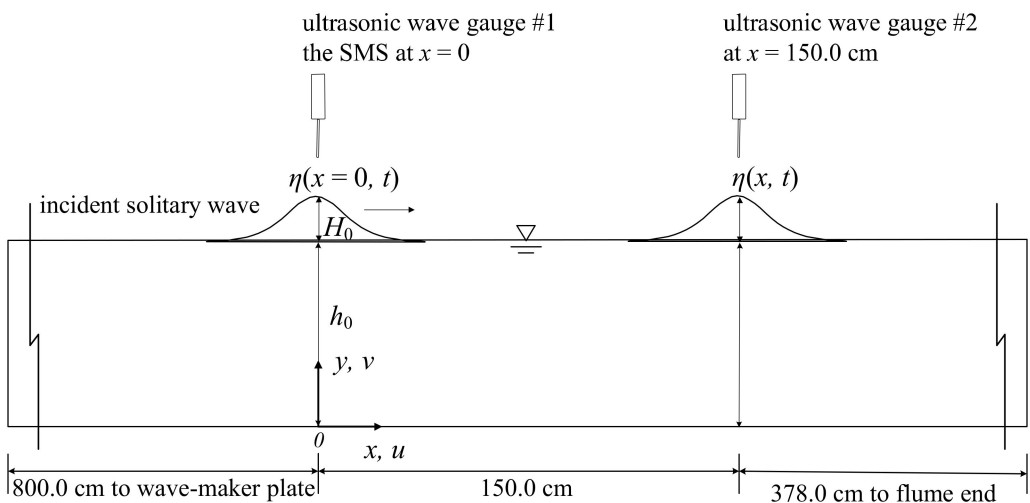

**Figure 1.** A schematic diagram illustrating a solitary wave traveling on a horizontal bed, together with the deployment of coordinate system and two wave gauges for measurement of the time elapse for the wave crest.

### 2.2. Deployment of Wave Gauges and HSPIV

Two ultrasonic wave gauges (Banner U-Gage S18U) were installed in the flume. One was at the SMS, used for measurements of the FSE $\eta(t)$ and $H_0$ for solitary waves. The other was at $x = 150.0$ cm, used to determine the time elapse for the wave crest to pass the two gauges and then estimate the wave celerity (see Figure 1).

Measurements of instantaneous velocity fields were made by an HSPIV system consisting of an argon-ion laser and a high-speed digital camera. A 5 W argon-ion laser (Coherent Innova-90) acted as a light source. A laser beam from the laser head was spread into a fan-shaped light sheet of about 1.5 mm thick, and then oriented upward through the bottom glass along the flume center line. Titanium dioxide particles, with a mean diameter of 1.8 μm, were uniformly seeded into the flow. The digital camera (Phantom VEO640, Vision Research), with a maximum 3200 Hz framing rate under a 1280 × 800-pixel resolution, recorded the instantaneous particle-laden images. Before performing the cross-correlation calculation for the instantaneous velocity field, the techniques of Laplacian edge-enhancement (Adrain and Westerweel [36]) and contrast enhancement (Cowen and Monismith [37]) were utilized to sharpen the edges and enhance the brightness of seeded particles in the images. The HSPIV algorithm permitted the instantaneous velocity field to be acquired from a pair of images, commencing at 64 × 64 pixel and terminating at 8 × 8 pixel.

### 2.3. Experimental Conditions

Two typical experimental cases were tested, case A: $h_0 = 8.0$ cm and $H_0 = 2.9$ cm and case B: $h_0 = 16.0$ cm and $H_0 = 5.8$ cm. They had different length scales; however, the same $H_0/h_0$ value (0.363). The two cases are utilized to provide a systematic comparison for time series of dimensionless FSEs, velocity components, and local and convective accelerations in both directions. It should be mentioned that the hydrodynamic features for case A were elucidated in Lin et al. [15] without discussing the similarity or FNS for different waves. Therefore, its results are used as a basis for comparison.

The fields of view (FOV) of the camera were set with different sizes for cases A and B with its center positioned at the SMS, as summarized in Table 1. The framing rate was fixed at 500 Hz for velocity measurements. For each case, a total of 20 repeated runs of the HSPIV measurements were performed for each case. To acquire the time histories of instantaneous velocity components at the SMS, a symmetric 11-point smoothing scheme with different weightings was utilized to remove noises in the velocity data. The ensemble-average

method was then used for the 20 repeated runs to attain the time series of ensemble-averaged horizontal and vertical velocities.

**Table 1.** A list of size, range, pixel resolution, and framing rate of each FOV for cases A and B.

| FOV | Size (cm × cm) | Range (cm × cm) | Pixel Resolution | Framing Rate | Case |
|---|---|---|---|---|---|
| FOVA | 16.80 × 16.80 | −8.40 ≤ x ≤ 8.40 | 1152 × 1152 | 500 | A |
| FOVB | 26.30 × 30.80 | −13.15 ≤ x ≤ 13.15 | 1152 × 1352 | 500 | B |

## 3. Preliminary Test

For case B, Figure 2 shows the ensemble-averaged velocity field for $2.0 \text{ cm} \leq y \leq 16.0 \text{ cm}$ in the vicinity of the SMS at $t = 0$. The measurement errors of these velocities are estimated by the mass flux method proposed by Chang and Liu [38] and adopted by Lin et al. [6,15].

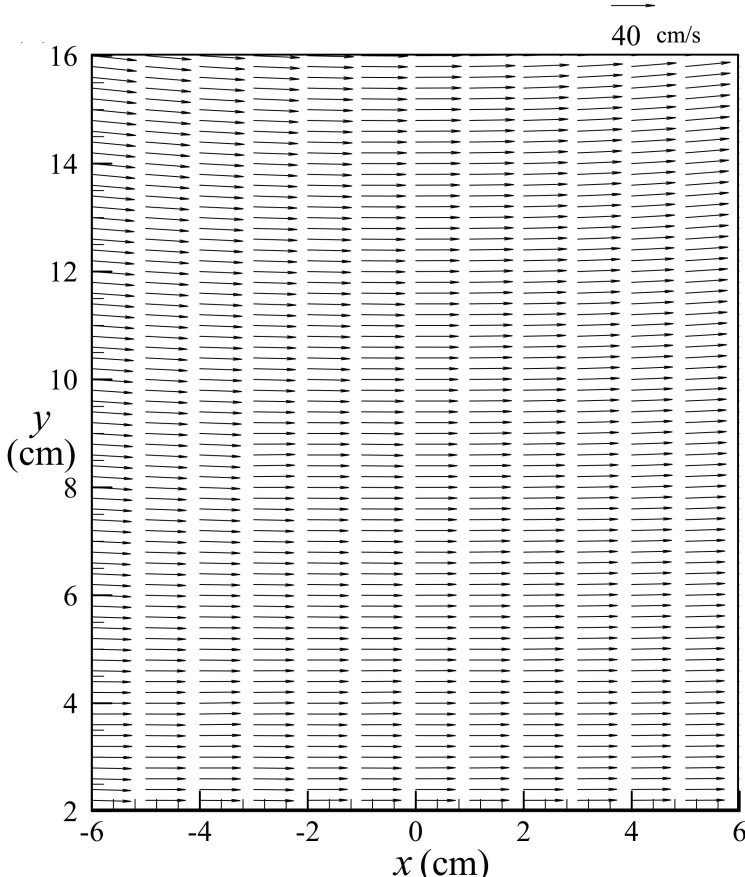

**Figure 2.** Velocity field beyond the near-bottom zone for $2.0 \text{ cm} \leq y \leq 16.0 \text{ cm}$ at $t = 0$ in the vicinity of the SMS (case B).

Based on the two-dimensional flow field, a check is made by computing the mass flux of each element, $M_f = |\partial u/\partial x + \partial v/\partial y| \times dA$. In the velocity field for case B, the element lengths and area are $\Delta x = \Delta y = 0.1827 \text{ cm}$ and $dA = \Delta x \times \Delta y = 0.0333 \text{ cm}^2$, respectively. A typical mass flux, $M_{f0}$ [$= (u_u)_{max} \times \Delta y$], is defined with $(u_u)_{max}$ being the maximum free-stream velocity beyond the bottom boundary layer (Lin et al. [8]) at $t = 0$. The relative error is denoted by $M_f/M_{f0}$. For most measurement positions without pronounced velocity gradient, the values of $M_f/M_{f0}$ values are below 2.5%. These small errors do validate high precision in the velocity measurements. As reported in Lin et al. [15], the $M_f/M_{f0}$ values for case A (with $1.0 \text{ cm} \leq y \leq 8.0 \text{ cm}$) are all below 2.8%. All these reconfirm the measurement accuracy of velocity field.

## 4. Description of Froude Number Similitude

As a solitary wave is one of the gravity waves (Dean and Dalrymple [39]), it is expected that the FNS (Daily and Harleman [40] and Munson et al. [41]) should exist for various external and internal flow properties of wave motion, which is dominated by gravity force. Therefore, in light of the FNS, two distinct solitary waves, however with identical $H_0/h_0$ value, as cases A and B, should have the same Froude number, $F_r$. In other words,

$$(F_r)_A = [U_s/(gL_s)^{1/2}]_A = [U_s/(gL_s)^{1/2}]_B = (F_r)_B \tag{1}$$

in which $L_s$ and $U_s$ are the representative length and velocity scales, respectively. For the incident solitary waves in this study, $L_s$ of case B is twice that of case A. The two cases satisfy the geometric similarity, with the ratio of two length scales expressed as:

$$(L_s)_B/(L_s)_A = (H_0)_B/(H_0)_A = (h_0)_B/(h_0)_A = 2.0 \tag{2}$$

Combining Equations (1) and (2), the ratios of velocity and time scales are

$$(U_s)_B/(U_s)_A = [(L_s)_B/(L_s)_A]^{1/2} = 2^{1/2} = 1.414 \tag{3}$$

$$(t_s)_B/(t_s)_A = [(L_s)_B/(U_s)_B]/[(L_s)_A/(U_s)_A] = [(H_0)_B/(H_0)_A]^{1/2} = 2^{1/2} = 1.414 \tag{4}$$

Using Equations (3) and (4), the ratio of acceleration scales is

$$(A_s)_B/(A_s)_A = [(U_s)_B/(t_s)_B]/[(U_s)_A/(t_s)_A] = [(U_s)_B/(U_s)_A] \times [(t_s)_A/(t_s)_B] = 1.0 \tag{5}$$

## 5. Results and Discussions

### 5.1. Time Series of FSE

For cases A and B, Figure 3 compares the time series of dimensionless FSE, $\eta(T)/H_0$, obtained at the SMS (i.e., $x/h_0 = 0$) with those predicted by the theoretical wave profile (Boussinesq [16] and Dean and Dalrymple [39]). The temporal variations in $\eta(T)/H_0$ agree well with the theoretical wave profile. Note that $H_0$ (= 5.8 cm) and $h_0$ (= 16.0 cm) for case B are twice those for case A ($H_0$ = 2.9 cm and $h_0$ = 8.0 cm). It turns out that a good agreement exists between the two cases, demonstrating *geometric similarity* in the relationship between $\eta(T)/H_0$ and $T$. Thus, the ratio of the two FSE scales is acquired from

$$[\eta_s/H_0]_B/[\eta_s/H_0]_A = [\eta_s]_B/[\eta_s]_A \times (L_s)_A/(L_s)_B = [\eta_s]_B/[\eta_s]_A \times (1/2) \approx 1.0 \tag{6}$$

Accordingly, the ratio of FSE scales is equal to

$$[\eta_s]_B/[\eta_s]_A = (H_0)_B/(H_0)_A = (L_s)_B/(L_s)_A \approx 2.0$$

thus justifying the geometric similarity between cases A and B (see Equation (2)).

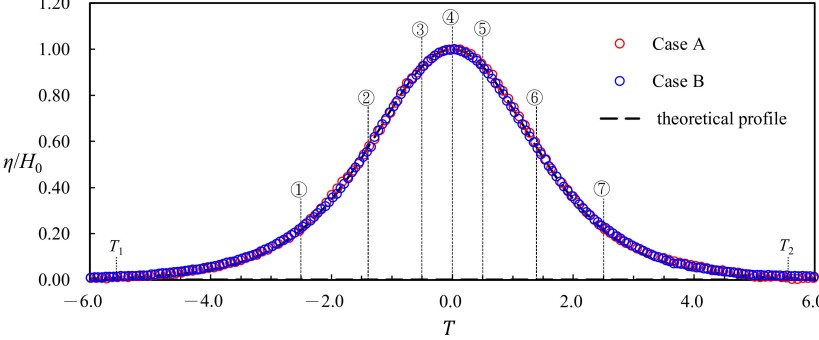

**Figure 3.** Comparison of the relationships between $\eta/H_0$ and $T$ for cases A and B, as well as that predicted by the theoretical wave profile at the SMS for $-6.00 \leq T \leq 6.00$. Note that the vertical lines circled with 1–7 correspond to $T$ = −2.50, −1.39, −0.50, 0, 0.50, 1.39, and 2.50, respectively.

### 5.2. Wave Celerity and Length

As indicated in Lin et al. [6], the measured values of wave celerity over horizontal bed can be obtained by $C_0' = \Delta x_w / \Delta t_w$, where $\Delta x_w$ (= 150.0 cm) is the spacing between the two gauges and $\Delta t_w$ is the time elapsed for a wave passing them. Data collection for both the measured data of $C_0'$ and the theoretical data of nonlinear wave celerity $C_0$ (= $[g \times (H_0 + h_0)]^{1/2}$) from Lin et al. [6,7,9,10] and Hwung et al. [42] is summarized in Table 2. It is found that $C_0' = (0.985–1.014) C_0 \approx C_0$, implying that $C_0'$ is well predicted by $C_0$ with a relative error $|(C_0' - C_0)/C_0|$ below 1.2%. Meanwhile, the measured $C_0'$ values for cases B and A are 144.0 and 102.0 cm/s, respectively. Namely, the ratio of the two measured wave celerities is

$$(C_0')_B/(C_0')_A = 1.412 \approx (2)^{1/2} = (C_0)_B/(C_0)_A = [(H_0 + h_0)^{1/2}]_B/[(H_0 + h_0)^{1/2}]_A \\ = [(L_s)_B/(L_s)_A]^{1/2} = (U_s)_B/(U_s)_A \tag{7}$$

as demonstrated by Equation (3). This fact confirms the validity of FNS in the celerity of distinct solitary waves with geometric similarity.

**Table 2.** Summary of the measured and predicted nonlinear wave celerities for solitary waves traveling over a horizontal bed (and then onto a sloping bed with the slope $S_0$). Note that the former measured data were all obtained in the same flume for the present and past studies.

| | $S_0$ | $H_0$ (cm) | $h_0$ (cm) | $H_0/h_0$ | $C_0'$ (cm/s) | $C_0$ (cm/s) | $C_0'/C_0$ |
|---|---|---|---|---|---|---|---|
| Present study | 0 | 2.90 (Case A) | 8.0 | 0.363 | 102.0 | 103.4 | 0.986 |
| | | 5.80 (Case B) | 16.0 | 0.363 | 144.0 | 146.2 | 0.985 |
| Lin et al. [6] | 1:10 | 1.33 | 10.0 | 0.133 | 104.9 | 105.4 | 0.995 |
| | | 1.23 | 7.0 | 0.176 | 90.1 | 89.9 | 1.003 |
| | | 1.48 | 7.0 | 0.211 | 90.9 | 91.2 | 0.997 |
| | | 2.10 | 8.02 | 0.262 | 98.1 | 99.6 | 0.985 |
| | | 1.93 | 7.02 | 0.275 | 93.5 | 93.7 | 0.998 |
| | | 2.87 | 8.0 | 0.359 | 102.0 | 103.3 | 0.988 |
| | | 2.69 | 7.0 | 0.384 | 99.0 | 97.7 | 1.013 |
| Lin et al. [7] | 1:5 | 2.90 | 8.0 | 0.363 | 102.0 | 103.4 | 0.986 |
| Lin et al. [9] | 1:3 | 2.90 | 8.0 | 0.363 | 102.0 | 103.4 | 0.986 |
| | | 2.10 | 8.0 | 0.263 | 98.2 | 99.5 | 0.987 |
| | | 2.74 | 16.0 | 0.171 | 134.1 | 135.6 | 0.988 |
| Lin et al. [10] | 1:3 | 2.90 | 8.0 | 0.363 | 102.2 | 103.4 | 0.988 |
| Hwung et al. [42] | 1:20 | 5.60 | 14.0 | 0.400 | 140.6 | 138.7 | 1.014 |

Moreover, as also seen in Figure 3, the dimensionless times $T_1$ and $T_2$ (corresponding to $t_1$ and $t_2$) identify the two instants at which the instantaneous FSEs are equal to 1.0% $H_0$, thus giving the representative periods of the two solitary waves (Liu et al. [20] and Sumer et al. [43]) of the solitary waves, $t_p$ (= $t_2 - t_1$). The wavelength $\lambda_0$ is determined by ($C_0 \times t_p$). As summarized in Table 3, $t_p = 1.004$ and 1.420 s and $\lambda_0 = 103.8$ and 207.7 cm for cases A and B, respectively. It is found that the relative wavelengths are both equal to $\lambda_0/h_0 = 12.98$, identical for cases A and B. Namely, the ratios of wavelengths and periods are

$$(\lambda_0)_B/(\lambda_0)_A = (L_s)_B/(L_s)_A = (h_0)_B/(h_0)_A = 2 \tag{8}$$

$$(t_p)_B/(t_p)_A = [(L_s)_B/(L_s)_A]^{1/2} = [(h_0)_B/(h_0)_A]^{1/2} = (2)^{1/2} = 1.414 \tag{9}$$

The two facts further demonstrate the geometric similarity of the two solitary waves, along with the ratio of wave periods, satisfying Equation (4) and following the FNS.

**Table 3.** Summary of the values of nonlinear wave celerity $C_0$, wave period $t_p$, and wavelength $\lambda_0$ for cases A and B.

| Case | $H_0$ (cm) | $H_0$ (cm) | $H_0/h_0$ | $C_0$ (cm/s) | $t_1$ (s) | $T_1$ | $t_2$ (s) | $T_2$ | $t_P$ (s) | $T_P$ | $\lambda_0$ (cm) | $\lambda_0/h_0$ |
|------|-----------|-----------|-----------|--------------|-----------|-------|-----------|-------|-----------|-------|------------------|-----------------|
| A | 2.90 | 8.0 | 0.363 | 103.4 | −0.502 | −5.56 | 0.502 | 5.56 | 1.004 | 11.12 | 103.8 | 12.98 |
| B | 5.80 | 16.0 | 0.363 | 146.2 | −0.710 | −5.56 | 0.710 | 5.56 | 1.420 | 11.12 | 207.7 | 12.98 |

## 6. Velocities

### 6.1. Time Series of Velocities

For cases A and B at the SMS, Figure 4a–c illustrate comparisons of the time series of dimensionless horizontal velocity $u(T)/C_0$ at $y/h_0 = 0.94$, 0.63, and 0.23, respectively. The trends are almost identical in both cases, confirming the similarity in the time series of $u(T)/C_0$ between them. It is interesting to note that, for $-6.00 \leq T < 0$ at the three measuring points, the magnitude of $u(T)/C_0$ increases from near zero to a maximum, suggestive of temporal acceleration of the flow. For $0 < T \leq 6.00$, it decreases from the maximum to virtually zero, indicative of temporal deceleration. At $T = 0$ for both cases, the maximum value of $u/C_0$ (= 0.283) at $y/h_0 = 0.94$ is larger than those (= 0.263 and 0.251) at $y/h_0 = 0.63$ and 0.23 for both cases. This trend indicates that the $u(T)/C_0$ distributions are non-uniform in the vertical direction. From these similarity results, the time series of $u(T)/C_0$ for $-6.00 \leq T < 6.00$ are identified to be symmetric about $T = 0$ for both cases, featuring an even-function shape with $u(T)/C_0 = u(-T)/C_0$.

It should be mentioned that, for $H_0/h_0 = 0.11$, 0.19 and 0.29, a comprehensive comparison of measured data for the FSEs as well as horizontal and vertical velocities of solitary waves with those predicted by distinct theories (including Boussinesq, McCowan, and Grimshaw theories) was reported by Lee et al. [24]. Due to the most frequent use of the Boussinesq theory, it would be interesting to make a comparison of velocity data obtained in the present study with those predicted by this theory. As also shown in Figure 4a–c for $y/h_0 = 0.94$, 0.63, and 0.23, the predicted values of horizontal velocity using nonlinear wave celerity $C_0 = [g \times (h_0 + H_0)]^{1/2}$ (in black dashed line) are mostly smaller than those of experimental data, especially around $T = -2.5$ and 2.5. Further, the predicted values around $T = 0$ either well match or slightly larger than the measured ones. Figure 4a–c also demonstrate the comparison of the predicted values of horizontal velocity employing linear wave celerity $C = (g \times h_0)^{1/2}$ (in green dashed line) with the measured data. It is surprisingly found that the overall trend and individual values of theoretical prediction are in very good agreement with those of the measured data. The fact strongly reflects that a fairly good prediction of the horizontal velocity can be achieved by Boussinesq theory if the linear wave celerity is incorporated in the calculation.

For cases A and B, Figure 5a–c show comparisons of the time series of dimensionless vertical velocity $v(T)/C_0$ at the SMS with $y/h_0 = 0.94$, 0.63, and 0.23, respectively. The temporal variations of $v(T)/C_0$ in both cases are almost identical, affirming similarity in the time series of $v(T)/C_0$. Based on the similarity results, the magnitudes of the positive and negative maxima (= 0.078 and −0.078) at $T = -1.39$ and 1.39 for $y/h_0 = 0.94$ (Figure 5a) are found to be 1.51 times those (= 0.0515 and −0.0515) for $y/h_0 = 0.63$ (Figure 5b), and 5.04 times those (= 0.0155 and −0.0155) for $y/h_0 = 0.23$ (Figure 5c). These results reveal the odd-function feature in the time series of $v(T)/C_0$ with respect to $T = 0$ (i.e., $v(T)/C_0 = -v(-T)/C_0$). Note that this characteristic is very distinct from those with asymmetric distributions as previously reported in Hsiao et al. [4] and Lee et al. [24]. Further, the results also highlight a prominent increase in the magnitude of $v(T)/C_0$ from bed to free surface at $T \neq 0$. For $T = 0$ at different heights, the values of $v(T)/C_0$ are virtually equal to zero for wave crest passing the SMS.

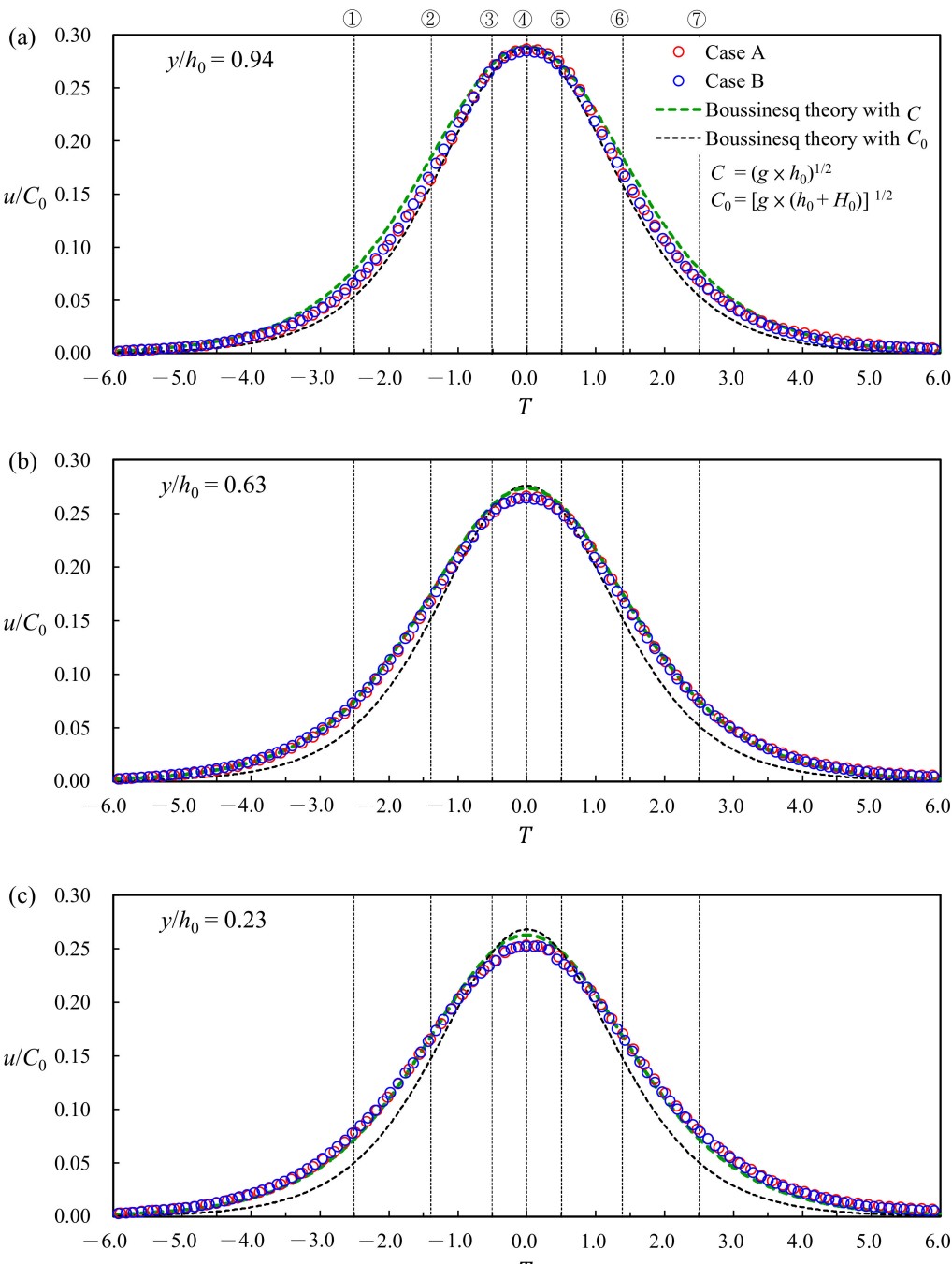

**Figure 4.** Comparisons of the time series of dimensionless horizontal velocities $u(T)/C_0$ for $y/h_0 =$ (**a**) 0.94; (**b**) 0.63; (**c**) 0.23 at the SMS for cases A and B. The vertical lines circled with 1–7 correspond to $T = -2.50, -1.39, -0.50, 0, 0.50, 1.39,$ and 2.50, respectively, as shown in Figure 2. Note that the predicted values of dimensionless horizontal velocity using Boussinesq theory with linear and nonlinear wave celerities are also shown in each subfigure.

As seen in Figure 5a,b for $y/h_0 = 0.94$ and 0.63, the magnitudes of the predicted values of vertical velocity using nonlinear wave celerity in Boussinesq theory (in black dashed line) are mostly less than those of experimental data, particularly for $-4.0 < T < -1.5$ and $1.5 < T < 4.0$. However, as evidenced in Figure 5c for $y/h_0 = 0.23$, due to fairly small magnitudes of vertical velocity closer to the horizontal bed, discernable distinction is hardly experienced between the predicted data and measured data. Figure 5a–c also shows the comparison of the predicted values of vertical velocity utilizing linear wave celerity (in

green dashed line) with the measured data. Similar to the results of horizontal velocity, the global trend and respective values of theoretical prediction are in good accordance with those of the measured data. The fact indicates that, if the linear wave celerity is used in the computation, Boussinesq theory well predicts the vertical velocity of a water particle.

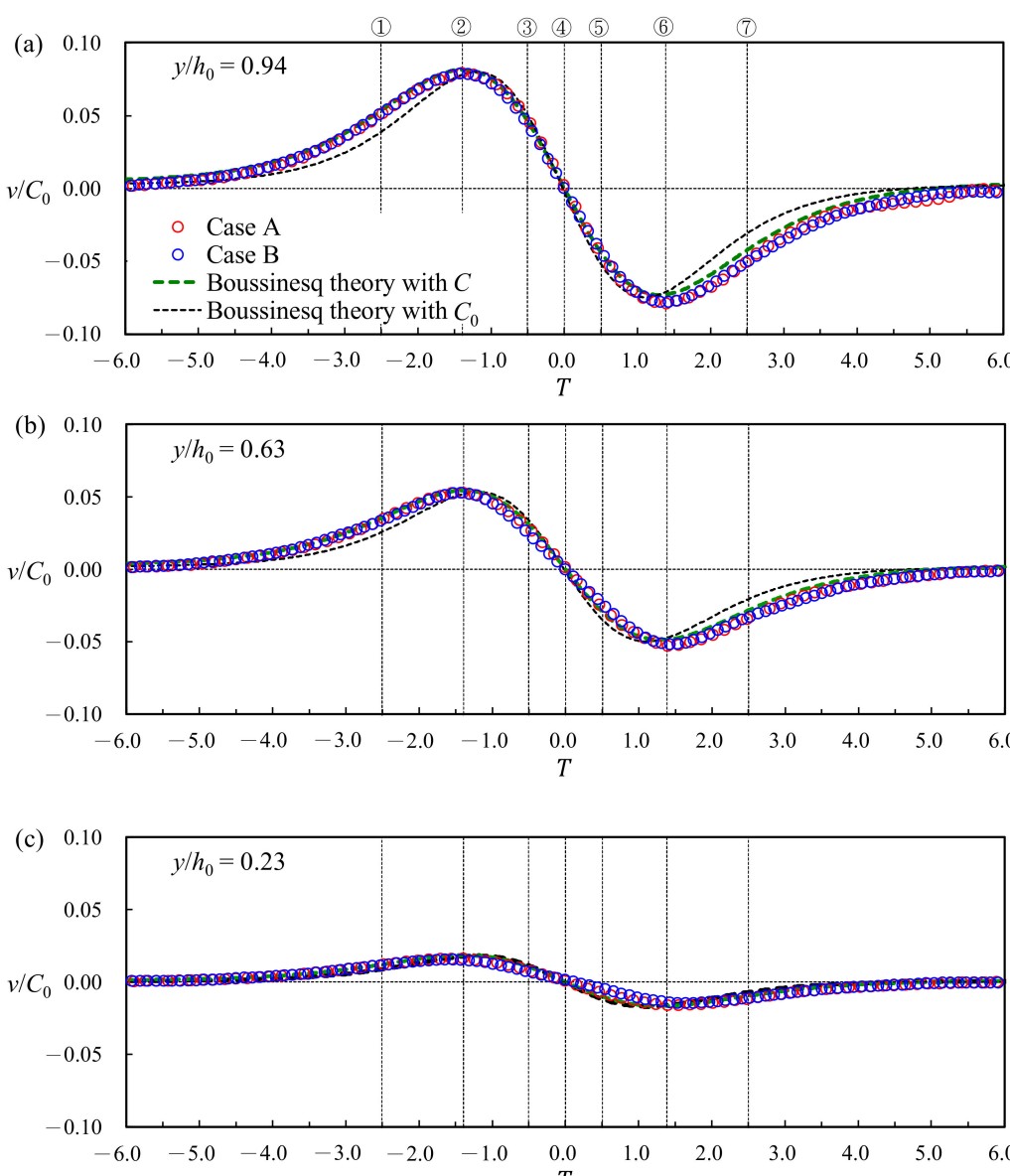

**Figure 5.** Comparisons of the time series of dimensionless vertical velocity $v(T)/C_0$ for $y/h_0 =$ (**a**) 0.94; (**b**) 0.63; (**c**) 0.23 at the SMS for cases A and B. The vertical lines circled with 1–7 correspond to $T = -2.50, -1.39, -0.50, 0, 0.50, 1.39$, and $2.50$, respectively. Note that the predicted values of dimensionless vertical velocity using Boussinesq theory with linear and nonlinear wave celerities are also shown in each subfigure.

Velocity Profiles

At the SMS for both cases, Figure 6 shows a comparison of dimensionless horizontal velocity profiles, $u(y/h_0, T)/C_0$, for $T = -2.50, -1.39, -0.50, 0, 0.50, 1.39$, and $2.50$. It is noted that, at any given $T$ value, the profile for case A is coincident with that for case B, illustrating that the similarity does exist between them. Besides, at $T = -0.50, -1.39$, and $-2.50$, they almost coincide with the respective counterparts at $T = 0.50, 1.39$, and $2.50$ for both cases. The truth reveals that similarity profiles appear in the dimensionless horizontal velocity distributions with reference to $T = 0$ at the SMS.

| | ① | ② | ③ | ④ | ⑤ | ⑥ | ⑦ |
|---|---|---|---|---|---|---|---|
| $T$ | $-2.50$ | $-1.39$ | $-0.50$ | 0 | 0.50 | 1.39 | 2.50 |
| Case A | ○ | ■ | ◆ | + | ⊞ | □ | ○ |
| Case B | ○ | ■ | ◆ | + | ⊞ | □ | ○ |

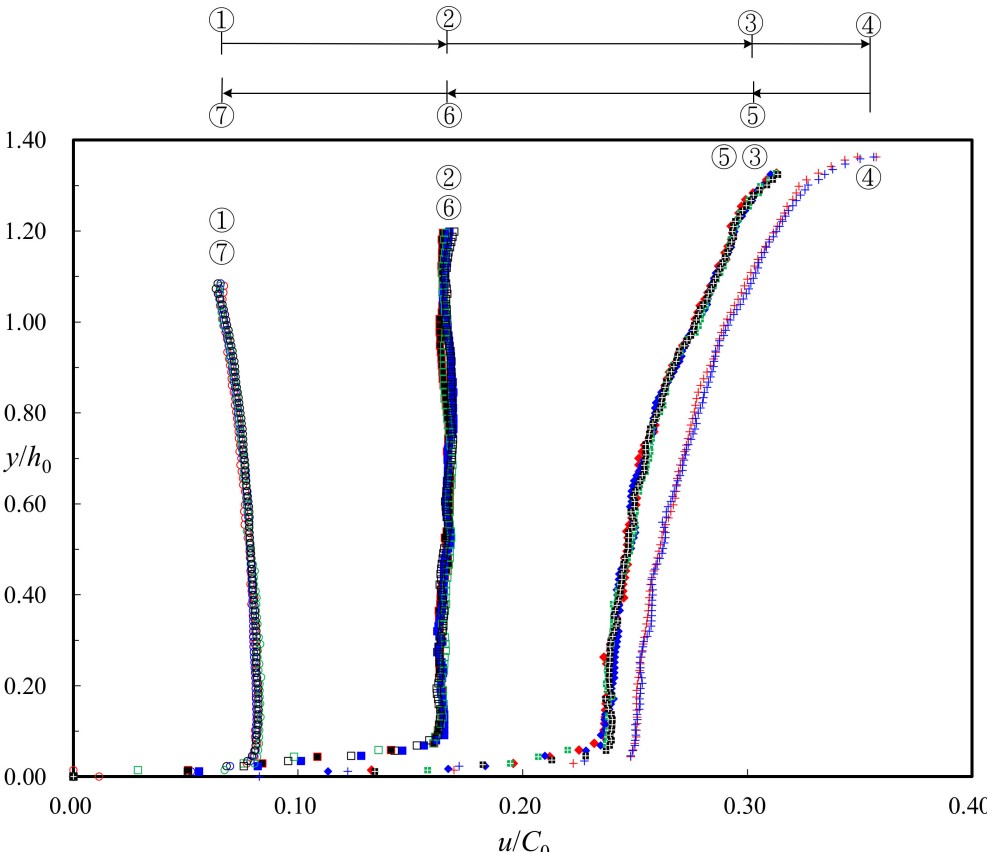

**Figure 6.** Comparison of the temporal variations in the $u(y/h_0)/C_0$ profiles for $-2.50 \leq T \leq 2.50$ at the SMS for cases A and B. Note that the circled numbers marked over the profiles are also shown in Figure 4a–c.

At the SMS, the temporal variations of dimensionless vertical velocity profiles, $v(y/h_0, T)/C_0$, are shown in Figure 7. At any $T$, each profile for case A well follows that for case B. This indicates the similarity in $v(y/h_0, T)/C_0$ between the two cases. Further, for both cases, the $v(T)/C_0$ profiles at $T = -0.50, -1.39$, and $-2.50$ almost collapse onto the respective counterparts at $T = 0.50, 1.39$, and $2.50$. These results indicate "symmetric" similarity profiles about $T = 0$ at the SMS. As observed for a given $y/h_0$ value (also see Figure 5a–c), $v(T)/C_0$ is positive and increases with $T$ for $-6.00 \leq T < -1.39$ (see ① and ② for $T = -2.50$ and $-1.39$). For $-1.39 < T \leq 0$, $v(T)/C_0$ decreases from the positive maximum at $T = -1.39$ down to zero at $T = 0$ (see ②–④ for $T = -1.39, -0.50$ and 0). It is also noted that, at any $T$, the $v(y/h_0)/C_0$ profile is characterized by a linearly increasing trend from bed to free surface, with the positive or negative maximum at $T = -1.39$ or 1.39 (② or ⑥).

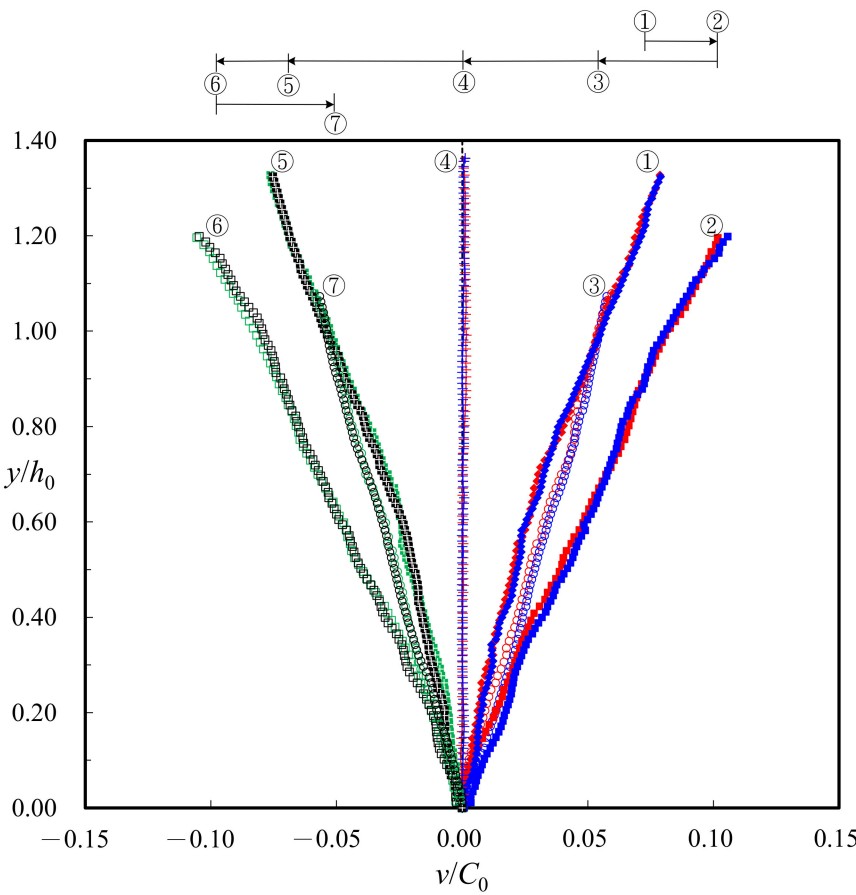

**Figure 7.** Comparison of the temporal variations in the $v(y/h_0)/C_0$ profiles for $-2.50 \leq T \leq 2.50$ at the SMS for cases A and B. The dimensionless times corresponding to the circled numbers (marked over the profiles) are also shown in Figure 5a–c.

For the maximum horizontal and vertical velocities, the ratios between cases A and B are expressed as:

$$
\begin{aligned}
(u_{\max}/C_0)_B/(u_{\max}/C_0)_A &= (u_{\max})_B/(u_{\max})_A \times [(H_0 + h_0)_A/(H_0 + h_0)_B]^{1/2} \\
&= (u_{\max})_B/(u_{\max})_A \times [(L_s)_A/(L_s)_B]^{1/2} = (u_{\max})_B/(u_{\max})_A \times (1/2)^{1/2} \approx 1.0
\end{aligned} \tag{10}
$$

and

$$
(v_{\max}/C_0)_B/(v_{\max}/C_0)_A = (v_{\max})_B/(v_{\max})_A \times [(L_s)_A/(L_s)_B]^{1/2} \approx 1.0 \tag{11}
$$

Combination of Equations (10) and (11) thus leads to

$$
(u_{\max})_B/(u_{\max})_A = (v_{\max})_B/(v_{\max})_A \approx 2^{1/2} = [(L_s)_B/(L_s)_A]^{1/2} = (U_s)_B/(U_s)_A \tag{12}
$$

which is identical to Equation (3) and follows the FNS.

By far, the experimental results have addressed the dimensionless parameters of not only time series of FSE, wave period, and wavelength, but also time series and profiles of velocity components. All of them demonstrate the similarity and FNS in the *geometric* and *kinematic features* for cases A and B. Similarities in the *hydrodynamic features*, in terms of local and convective accelerations, are discussed below.

### 6.2. Local Accelerations

The local accelerations in the horizontal and vertical directions are represented by $A_{1x}(x, y, t) = \partial u(x, y, t)/\partial t$ and $A_{1y}(x, y, t) = \partial v(x, y, t)/\partial t$, respectively. It is known that the time interval $\Delta t_{acce}$ employed in the calculations should be as small as possible. From a practical viewpoint, to obtain the values of $A_{1x}$ or $A_{1y}$ is, however, difficult due to rapid

temporal fluctuations in the image-based HSPIV system. In other words, even the use of distinct small time intervals in the differential computation does result in large differences in local accelerations, which is evidenced in Lin et al. [10,12,15] and Jensen et al. [29]. A series of failures in the convergent tests were addressed in their studies. An alternative method is to obtain an appropriate range of the neighboring time intervals, in which the relative bias of each calculated result of the local acceleration is below 4.0% of the averaged value is used. Detailed procedure with trial-and-error for computing local accelerations is referred to in Lin et al. [10,12,15].

In the present study, the appropriate $\Delta t_{acce}$ ranges between 0.016 and 0.032 s for case A or between 0.022 and 0.046 s for case B. The most suitable is identified to be $(\Delta t_{acce})_A = 0.024$ s or $(\Delta t_{acce})_B = 0.034$ s, which is 12 or 17 times the framing time interval $\Delta t_{framing}$ (= $1/500$ s = 0.002 s, see Table 1) for case A or B. Note that the two choices are in accordance with those of Lin et al. [10,12,15] and Jensen et al. [29], with all showing $\Delta t_{acce}$ is much larger than $\Delta t_{framing}$, but not as small as $\Delta t_{framing}$. Together with the use of a central difference scheme, the local accelerations are acquired as $A_{1x} \approx \Delta u / \Delta t_{acce}$ and $A_{1y} \approx \Delta v / \Delta t_{acce}$. Similar to the approach in Jensen et al. [29], the time histories of local accelerations were processed by a symmetric seven-point smoothing scheme.

6.2.1. Times Series of Local Accelerations

For both cases, Figure 8a–c compare the time series of $A_{1x}/g$ at $(x/h_0, y/h_0) = (0, 0.94)$, $(0, 0.63)$, and $(0, 0.23)$, respectively. The results clearly show that the two $A_{1x}/g$ trends are almost identical, thus demonstrating the similarity in temporal variations of $A_{1x}(T)/g$. From the similarity results shown in Figure 4a–c for $T < 0$, the $u(t)$ magnitude increases with increasing $T$. This indicates that the flow accelerates temporally at the SMS. For $T > 0$, it decreases with increasing $T$, highlighting that the flow decelerates temporally. Accordingly, the values of $A_{1x}/g$ are positive for $-6.00 \leq T < 0$ and negative for $0 < T \leq 6.00$, along with $A_{1x}/g$ equal to zero for $T = 0$. Further, for cases A and B, the almost identical positive or negative $A_{1x}/g$ maxima, i.e., $A_{1x+}/g$ or $A_{1x-}/g$, appear at $T = -1.39$ or $T = 1.39$, thus resulting in $(A_{1x+})_A/g \approx (A_{1x+})_B/g \approx -(A_{1x-})_A/g \approx -(A_{1x-})_B/g$ as seen in Figure 8a–c. Note that the magnitudes of the $A_{1x}/g$ maxima (= 0.142 or 0.143) at $y/h_0 = 0.94$ are larger than those (= 0.118 or 0.117) at $y/h_0 = 0.63$ and those (= 0.108 or 0.111) at $y/h_0 = 0.23$.

Moreover, Figure 9a–c compare the time series of $A_{1y}(T)/g$ for $(x/h_0, y/h_0) = (0, 0.94)$, $(0, 0.63)$, and $(0, 0.23)$, respectively. Similar to the changes of $A_{1x}/g$ in Figure 8a–c, the results reveal that the data trends of $A_{1y}/g$ overlap with insignificant discrepancy, confirming that similarity exists in the time series of $A_{1y}(T)/g$ for both cases. It is found that the values of $A_{1y}/g$ for both cases are positive for $-6.00 \leq T < -1.39$ and $1.39 < T \leq 6.00$ and negative for $-1.39 < T < 1.39$. In addition, note that, for cases A and B, $A_{1y}/g = 0$ occurs for $T = -1.39$ and 1.39, at which $A_{1x}/g$ exhibits the respective maxima. In addition, the magnitudes of the mean positive and negative maxima of $A_{1y}(T)/g$ (= 0.038 or 0.108) at $y/h_0 = 0.94$ are larger than those (= 0.026 or 0.067) at $y/h_0 = 0.63$ and those (= 0.006 or 0.016) at $y/h_0 = 0.23$ for both cases.

| | ⓐ | ⓑ | ⓒ | ⓓ | ⓔ | ⓕ | ⓖ | ⓗ | ⓘ | ⓙ | ⓚ | ⓛ | ⓜ | ⓝ | ⓞ |
|---|---|---|---|---|---|---|---|---|---|---|---|---|---|---|---|
| $T$ | −5.50 | −3.50 | −2.50 | −2.00 | −1.39 | −0.90 | −0.50 | 0 | 0.50 | 0.90 | 1.39 | 2.00 | 2.50 | 3.50 | 5.50 |
| Case A | ⊟ | ⊖ | ○ | ◈ | ■ | ◑ | ◆ | + | ⊞ | △ | □ | ◈ | ○ | ◓ | ▲ |
| Case B | ⊟ | ⊖ | ○ | ◈ | ■ | ◑ | ◆ | + | ⊞ | △ | □ | ◈ | ○ | ◓ | ▲ |

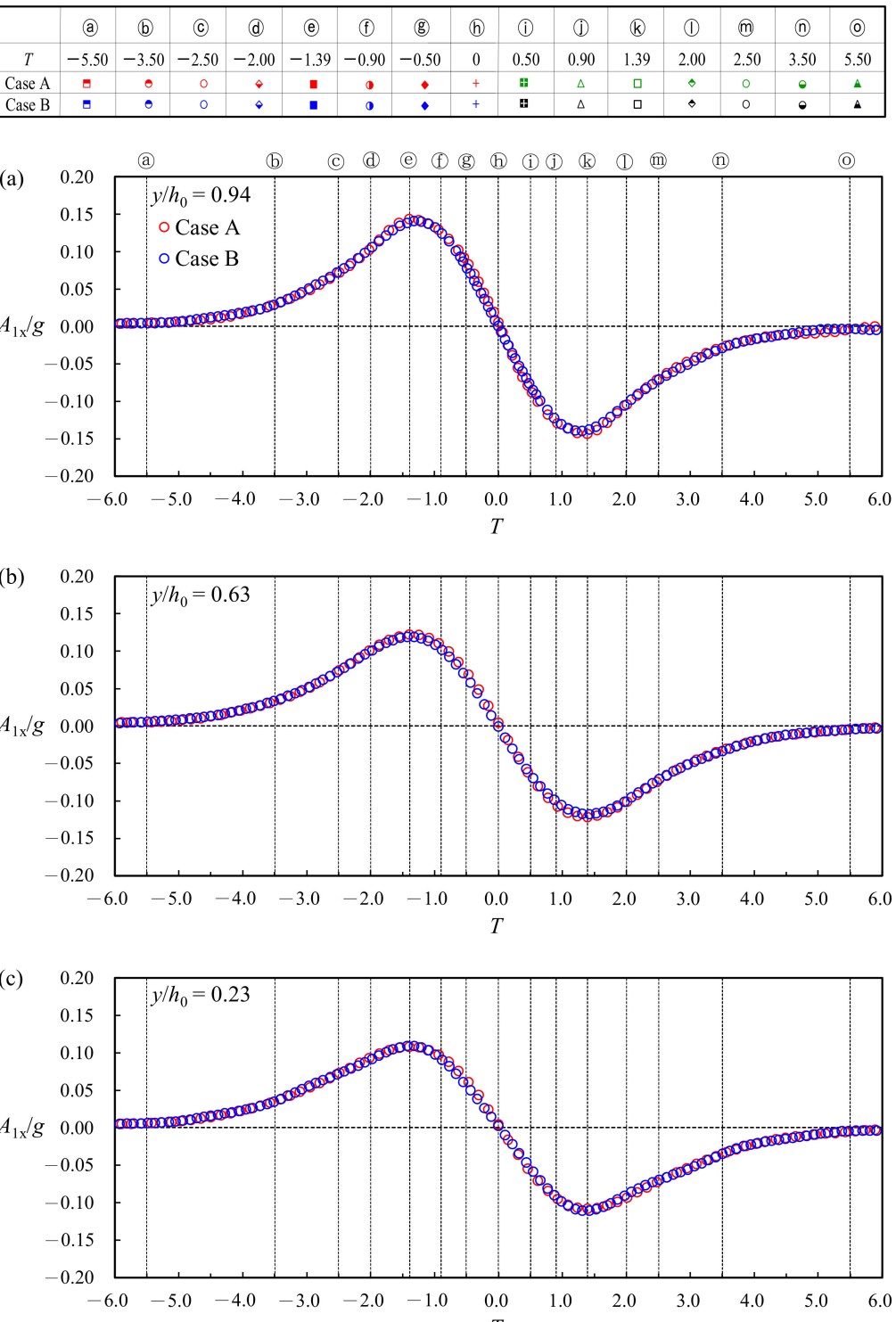

**Figure 8.** Comparisons of the time series of $A_{1x}(T)/g$ for $y/h_0$ = (**a**) 0.94; (**b**) 0.63; (**c**) 0.23 at the SMS for cases A and B.

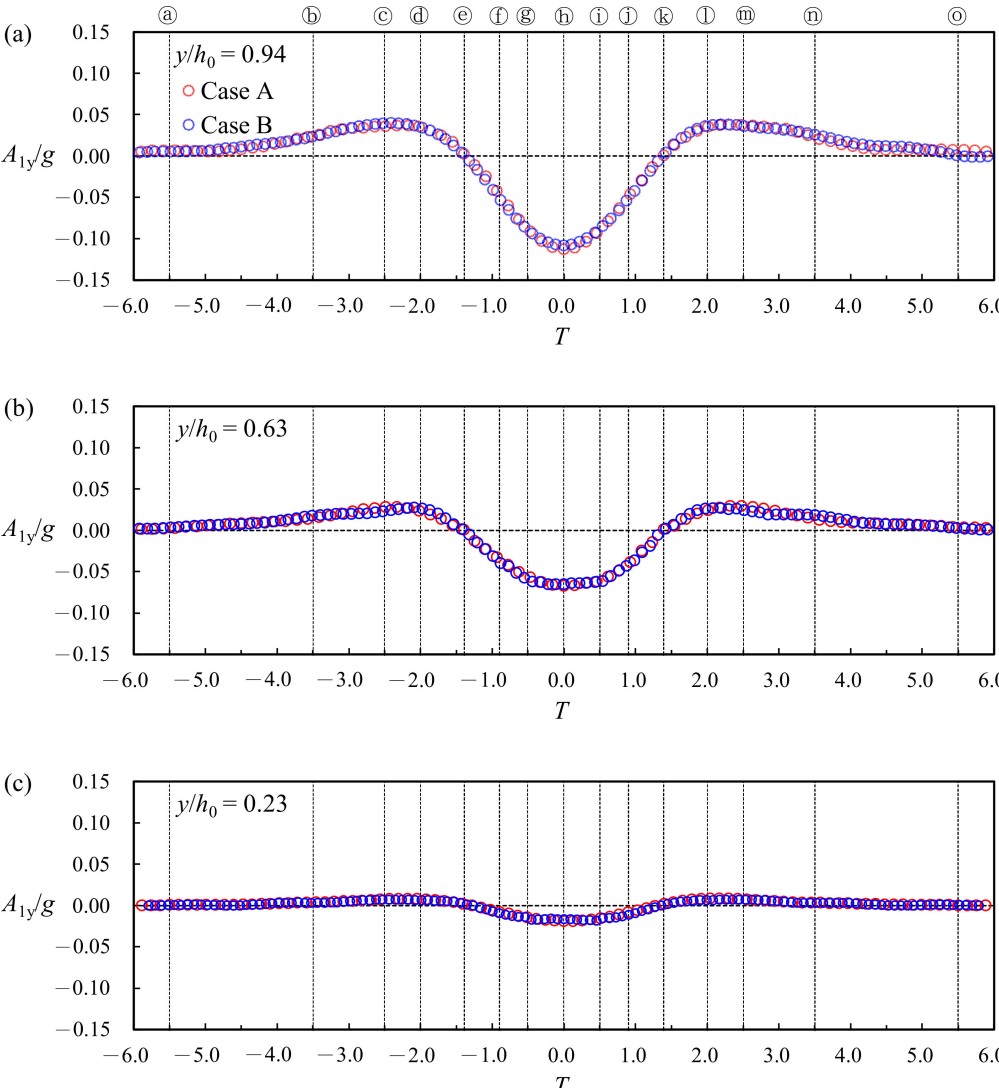

**Figure 9.** Comparisons of the time series of $A_{1y}(T)/g$ for $y/h_0$ = (**a**) 0.94; (**b**) 0.63; (**c**) 0.23 at the SMS for cases A and B. The dimensionless times correspond to the circled a–o marked over each subfigure are shown in the sub-table of Figure 8.

### 6.2.2. Profiles of Local Accelerations

For both cases at the SMS, Figures 10 and 11 illustrate, in both directions, a series of comparisons for the profiles of dimensionless local accelerations $A_{1x}(y/h_0, T)/g$ and $A_{1y}(y/h_0, T)/g$ with $T$ varying from −5.50 to 5.50. The results show that, for a given $T$, the profile in case A almost collapses onto that in case B. This evidence supports the similarity in the profiles of $A_{1x}/g$ and $A_{1y}/g$ for both cases. Further, the ratios of their respective maxima can be written as:

$$(A_{1x+})_B/(A_{1x+})_A = (A_{1x^-})_B/(A_{1x^-})_A = (A_s)_B/(A_s)_A \approx 1 \tag{13}$$

$$(A_{1y+})_B/(A_{1y+})_A = (A_{1y^-})_B/(A_{1y^-})_A = (A_s)_B/(A_s)_A \approx 1 \tag{14}$$

The equations confirm that, in either direction between the two cases, the ratios of local acceleration scales are equal to unity, which are identical to Equation (5) and satisfies the FNS.

| | ⓐ | ⓑ | ⓒ | ⓓ | ⓔ | ⓕ | ⓖ | ⓗ | ⓘ | ⓙ | ⓚ | ⓛ | ⓜ | ⓝ | ⓞ |
|---|---|---|---|---|---|---|---|---|---|---|---|---|---|---|---|
| $T$ | −5.50 | −3.50 | −2.50 | −2.00 | −1.39 | −0.90 | −0.50 | 0 | 0.50 | 0.90 | 1.39 | 2.00 | 2.50 | 3.50 | 5.50 |
| Case A | | | | | | | | | | | | | | | |
| Case B | | | | | | | | | | | | | | | |

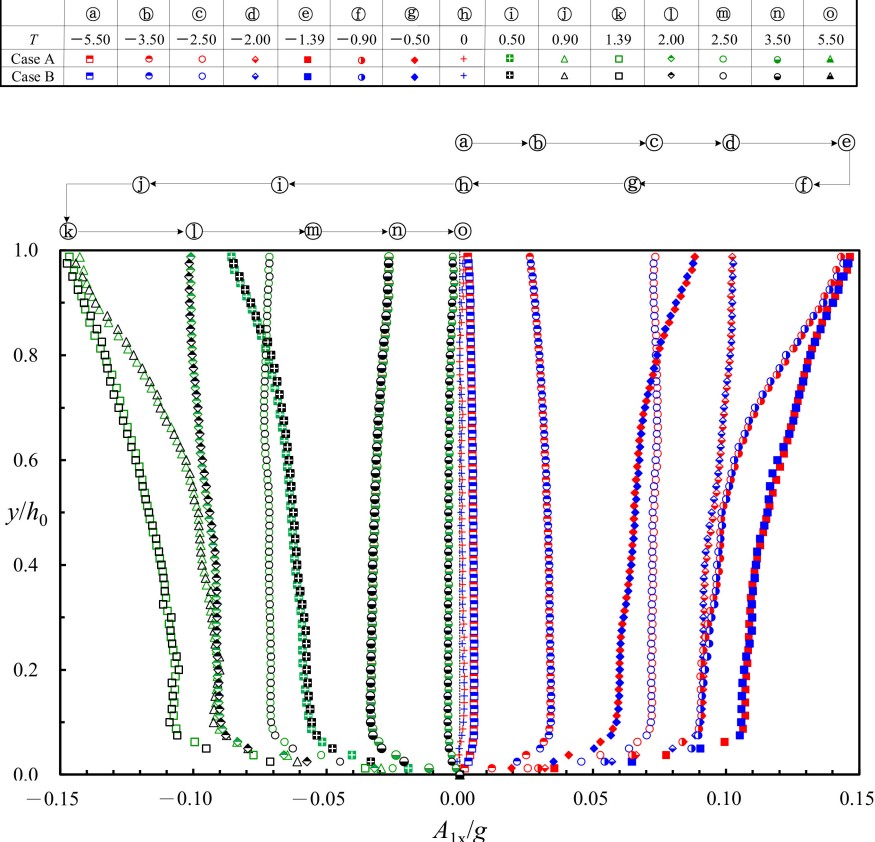

**Figure 10.** Comparisons of the profiles of dimensionless local acceleration in the horizontal direction, $A_{1x}/g$, at the SMS for cases A and B.

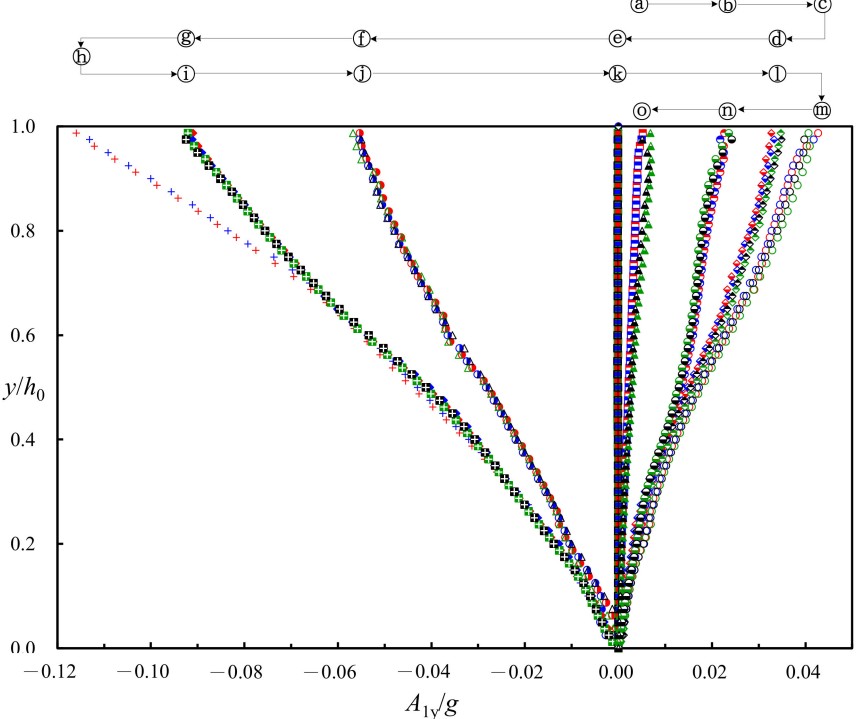

**Figure 11.** Comparisons of the profiles of dimensionless local acceleration in the vertical direction, $A_{1y}/g$, at $x/h_0 = 0$ for cases A and B. The dimensionless times correspond to circled a–o marked over the figure are shown in the sub-table of Figure 10.

As seen in Figure 10 for both cases, the temporal variations in the $A_{1x}(y/h_0, T)/g$ profiles are symmetric about $T = 0$ and characterized by an even-function shape. This is attributable to the symmetric feature in time series of $u/C_0$ (see Figure 4a–c), thus rendering the magnitudes of $A_{1x}/g$ for $T < 0$ almost equal to those for $T > 0$ (but with opposite signs). At distinct $y/h_0$ values for cases A and B, the negative and positive maxima in the $A_{1x}/g$ profiles appear at $T = -1.39$ and 1.39, respectively. Due to the odd-function shape in time series of $v/C_0$ (see Figure 5a–c), temporal change in the $A_{1y}/g$ profiles (Figure 11) shows a fairly distinct trend from that of $A_{1x}/g$. Note that, in either case, the values in each $A_{1y}/g$ profile at a given $|T|$ for $-6.00 \leq T < 0$ are nearly equal to those for $0 < T \leq 6.00$, with the positive maxima at $|T| = 2.50$.

### 6.3. Convective Accelerations

The convective accelerations in the horizontal and vertical directions are defined as $A_{2x}(x, y, t) = u \times \partial u/\partial x + v \times \partial u/\partial y$ and $A_{2y}(x, y, t) = u \times \partial v/\partial x + v \times \partial v/\partial y$, respectively. As indicated in Lin et al. [10,12,15], the use of distinct minute spatial intervals in differential calculations leads to significant differences in convective acceleration, which results in the failure to make reasonable estimations. An alternative approach is used to find, by trial-and-error, the appropriate range of neighboring spatial intervals, in which the relative bias for each calculation result varies only within 4.0% of the averaged value. Details of the convective acceleration calculations are found in Lin et al. [10,12,15].

It is worth noting that the smallest element sizes for the velocity fields obtained by HSPIV are $(\Delta x \times \Delta y) = (\Delta x)^2 = (\Delta y)^2 = (0.1167 \text{ cm})^2$ for case A and $(0.1827 \text{ cm})^2$ for case B. After a series of tests, the appropriate spatial intervals used to calculate $A_{2x}$ and $A_{2y}$ range $(8–20) \Delta x$ for case A and $(6–13) \Delta x$ for case B. The most proper ones are $(\Delta x_{acce})_A = (\Delta y_{acce})_A = 17 \times 0.1167 \text{ cm} = 1.98 \text{ cm}$ for case A and $(\Delta x_{acce})_B = (\Delta y_{acce})_B = 11 \times 0.1827 \text{ cm} = 2.01 \text{ cm}$ for case B. Along with the use of central difference scheme, $A_{2x}(t)$ and $A_{2y}(t)$ are attained by $[u \times \Delta u/\Delta x_{acce} + v \times \Delta u/\Delta y_{acce}]$ and $[u \times \Delta v/\Delta x_{acce} + v \times \Delta v/\Delta y_{acce}]$, respectively. Finally, a symmetric seven-point smoothing scheme with distinct weightings removed the noises in the time histories of convective acceleration.

As reported in Lin et al. [12,15], the convective acceleration maxima in both directions are much smaller than the *local* acceleration ones. Besides, the magnitude of $([A_{2x}(y)]^2 + [A_{2y}(y)]^2)^{1/2}/g$ increases from the bed to the free surface, revealing that $A_{2x}(y)$ and $A_{2y}(y)$ are functions of $y/h_0$ with complex situations. For simplicity, the depth-averaged method is used to acquire the representative result.

For both cases at the SMS, Figure 12 shows the temporal variations of dimensionless depth-averaged (da) convective accelerations in the horizontal direction $[A_{2x}(T)]_{da}/g$. The magnitudes of negative or positive maxima in $(A_{2x}(T))_{da}/g$ (= $-0.022$ or $0.021$) appear at $T = -1.09$ or $T = 1.09$, slightly greater than those (= $-0.018$ or $0.018$) at $T = -1.39$ or $T = 1.39$. Similar to the $A_{1x}(T)/g$ and $A_{1y}(T)/g$ features in Figures 8a–c and 9a–c, the results in Figure 12 show that the change of $[A_{2x}(T)]_{da}/g$ in case A nearly coincides with that in case B. This confirms the similarity in the time series of $[A_{2x}(T)]_{da}/g$ between these two cases. Therefore, in the horizontal direction, the ratio of the depth-averaged convective acceleration scales is expressed as

$$([A_{2x}]_{da})_{s,B}/([A_{2x}]_{da})_{s,A} \approx 1.0 \tag{15}$$

which is equivalent to Equation (5) following the FNS.

| | ⓐ | ⓑ | ⓒ | ⓓ | ⓔ | ⓕ | ⓖ | ⓗ | ⓘ | ⓙ | ⓚ | ⓛ | ⓜ | ⓝ | ⓞ |
|---|---|---|---|---|---|---|---|---|---|---|---|---|---|---|---|
| $T$ | $-5.50$ | $-3.50$ | $-2.50$ | $-2.00$ | $-1.39$ | $-0.90$ | $-0.50$ | $0$ | $0.50$ | $0.90$ | $1.39$ | $2.00$ | $2.50$ | $3.50$ | $5.50$ |

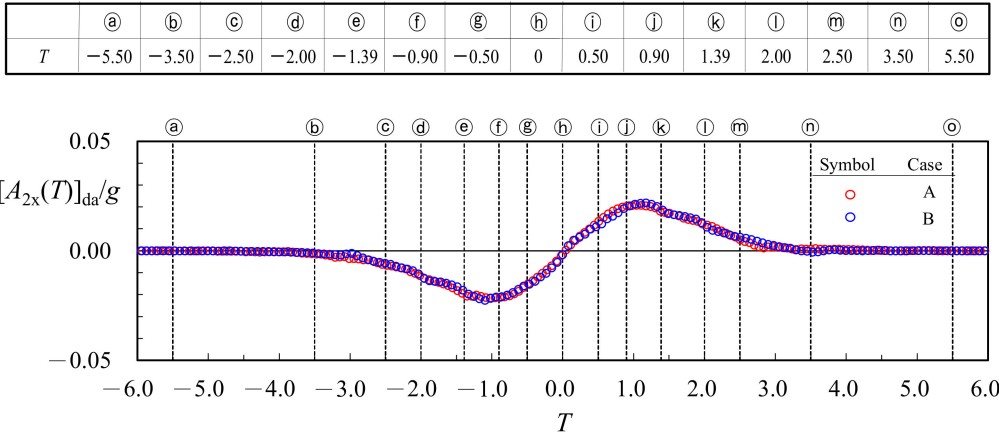

**Figure 12.** Comparison of temporal variations in the dimensionless convective acceleration in the horizontal direction, $[A_{2x}(T)]_{da}/g$, at the SMS for cases A and B. Note that the scale of ordinate is different from those in Figures 8 and 9.

Finally, Figure 13 compares the temporal variations of dimensionless depth-averaged convective acceleration in the vertical direction, $[A_{2y}(T)]_{da}/g$. It is obvious that the differences in $[A_{2y}(T)]_{da}/g$ are negligible between the two cases, affirming the similarity in the time series.

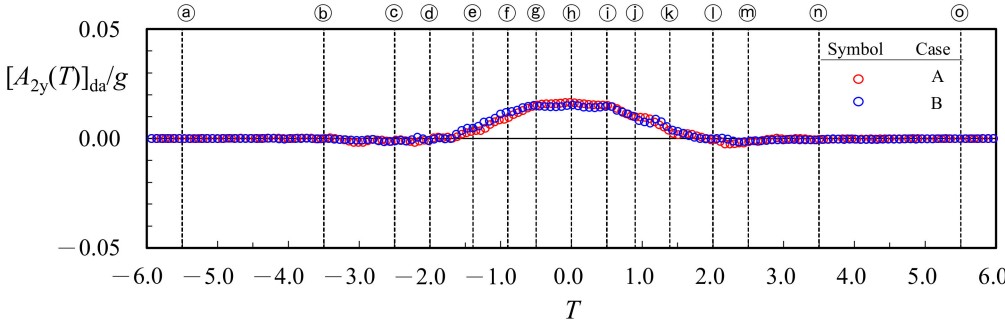

**Figure 13.** Comparison of the temporal variations in dimensionless convection acceleration in the vertical direction, $[A_{2y}(T)]_{da}/g$, at the SMS for cases A and B. Note that the scale of ordinate is different from those in Figures 8 and 9.

The ratio of the depth-averaged convective acceleration scales in the vertical direction is expressed as

$$([A_{2y}]_{da})_{s,B}/([A_{2y}]_{da})_{s,A} \approx 1.0 \tag{16}$$

Equation (16) is equal to Equation (5), which satisfies the FNS. Note that, in either case, the maximum magnitudes of $[A_{2y}]_{da}/g$ amount to 0.014–0.015, slightly smaller than those of $[A_{2x}]_{da}/g$ (= 0.021–0.022).

## 7. Conclusions

A pair of experiments were performed for two solitary waves, with distinct length scales, however identical $H_0/h_0$ value (0.363), propagating over a horizontal bed. The main purpose is to demonstrate that similarity and FNS are valid in the kinematic and hydrodynamic features of the two solitary waves. The findings of this study are summarized as follows.

1.  In either direction, similarity and FNS hold true for the time series of dimensionless FSEs, velocity components, and local and convective accelerations.
2.  The similarity and FNS are also valid for dimensionless wave celerity and wavelength, horizontal and vertical velocity profiles, and local acceleration profiles.

3.　All the similarities and FNSs demonstrate that gravity force is the most significant factor that dominates flow kinematics and hydrodynamics of solitary waves. The fact implies that, even for small-scale experiments with $h_0 \geq 8.0$ cm, viscous friction and surface tension play negligible roles in affecting FSE, flow velocity, and acceleration.

4.　Based on the comparisons made between velocity data obtained in this study and predicted values, Boussinesq theory is found to well predict the horizontal and vertical velocities of a water particle if the linear wave celerity is incorporated into the calculation.

5.　The *horizontal velocity* is nonuniform in the vertical direction, particularly near the free surface. This feature is contrary to the traditional uniform distribution well recognized in the past. Its time series are symmetric for $-6.00 \leq T < 6.00$ (featuring an even-function shape) about $T = 0$, at which the maximum horizontal velocity occurs.

6.　As pre-passing (post-passing) of the wave crest is accompanied by ascending (descending) FSE, the *vertical velocity* is positive (negative). Their respective maximum of equal magnitude in the time series of vertical velocity appears at $T = -1.39$ and $1.39$, with an odd-function distribution about $T = 0$. This feature is different from those with asymmetric distributions as previously reported. The vertical velocity increases linearly from zero at the bed to a temporally-varied maximum at the free surface for $T \neq 0$. For $T = 0$, however, it is equal to zero at different heights.

7.　In the horizontal direction, the temporal variation in time series (profiles) of *local acceleration* is characterized by an odd-function (even-function) shape about $T = 0$. The positive and negative maxima take place at $T = -1.39$ and $1.39$, respectively, at distinct $y/h_0$ values.

8.　In the vertical direction, an even-function shape features the temporal variation in the time series of *local acceleration*. The profiles with negative and positive maxima appear for $T = 0$ and $|T| = 2.50$, respectively, with virtually zero values at distinct heights for $T = -1.39$ and $1.39$.

9.　In either direction, the magnitudes of positive and negative maxima in the time series of depth-averaged *convective acceleration* are much smaller than those of the local acceleration.

**Author Contributions:** C.L. was responsible for project administration, technical supervision and quality control of experimental results, and funding acquisition. Execution of the experimental tests, image processing, free surface, velocity analyses, and calculation of the accelerations were performed by M.-J.K. and S.-C.H. The manuscript was written by C.L. Manuscript modifications and corrections were completed by J.Y., R.V.R. and J.-M.Y. All authors have read and agreed to the published version of the manuscript.

**Funding:** This research was supported by the Ministry of Science and Technology, Taiwan via Grant Nos. MOST 108-2221-E-005-015-MY3 and MOST 109-2221-E-005-026-MY3 to Department of Civil Engineering, National Chung Hsing University, Taichung, Taiwan; and MOST 108-2115-M-126-003 and MOST 109-2115-M-126-002 to Department of Data Science and Big Data Analytics, Providence University, Taichung, Taiwan. This study was also partially supported by Royal Institute of Technology (KTH) and the Swedish Hydropower Centre (SVC), Stockholm, Sweden.

**Institutional Review Board Statement:** Not applicable.

**Informed Consent Statement:** Not applicable.

**Data Availability Statement:** The data that support the findings of this study are available from the corresponding author upon reasonable request.

**Acknowledgments:** Special thanks to UTOPIA Instruments Co., Ltd. for helping the installation and testing the high-speed digital camera used. The authors are grateful to Po-Yu Chuang, Jie-Ming Syu, and Wei-Chih Pan for HSPIV measurements and data analysis at Fluid Mechanics Laboratory, Department of Civil Engineering, NCHU.

**Conflicts of Interest:** The authors declare no conflict of interest.

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
