# Peer review of "Similarity and Froude Number Similitude in Kinematic and Hydrodynamic Features of Solitary Waves over Horizontal Bed"

_processes, doi:10.3390/pr9081420_

Round 1

Reviewer 1 Report

Experimental work 'Similarity and Froude number similitude in kinematic and hydrodynamic features of solitary waves over horizontal bed' devoted to the determination of the Froude numbers for two solitary waves is certainly of interest in the study of solitons in shallow water. The results obtained by the authors make it possible to gain a deeper understanding of the kinematics and hydrodynamics of such nonlinear phenomena. I believe that the obtained measurement results were carried out with minimal errors and are trustworthy.

In my opinion, it would be good for authors to add a few sentences about the coincidence of experimental results this work and theoretical predictions from references.

Notes: The authors provide links to articles that analytically examine such issues. Unfortunately, they (the authors) do not make comparisons with theoretical results.

Author Response

Point 1: Experimental work 'Similarity and Froude number similitude in kinematic and hydrodynamic features of solitary waves over horizontal bed' devoted to the determination of the Froude numbers for two solitary waves is certainly of interest in the study of solitons in shallow water. The results obtained by the authors make it possible to gain a deeper understanding of the kinematics and hydrodynamics of such nonlinear phenomena. I believe that the obtained measurement results were carried out with minimal errors and are trustworthy.

Response 1:   Thanks so much for your support and encouragement!

Point 2: In my opinion, it would be good for authors to add a few sentences about the coincidence of experimental results this work and theoretical predictions from references. 

Notes: The authors provide links to articles that analytically examine such issues. Unfortunately, they (the authors) do not make comparisons with theoretical results.

Response 2: Thank for providing this valuable review comment. We have added the comparisons of the experimental velocity data with those of Boussinesq theory. The supplements have been addressed in Lines 259-282 and 296-314 in the revised manuscript.  Thanks so much!

Finally, the authors would like to express their deepest appreciation to the Reviewer for providing the valuable review comment. 

Reviewer 2 Report

The paper is recommended to publish.

Author Response

Point 1: The paper is recommended to publish.  

Response 1:   Thanks so much for your support and encouragement!